# Categories vs semantic features: What shapes the similarities people discern in photographs of objects?

Siddharth Suresh[1,2,3], Wei-Chun Huang[2], Kushin Mukherjee[1,3], and Timothy T. Rogers[1,3]

[1]Department of Psychology, University of Wisconsin-Madison, Madison, USA.
[2]Department of Computer Science, University of Wisconsin-Madison, Madison, USA. Email:
`siddharth.suresh@wisc.edu`
[3]Wisconsin Institute of Disscovery, University of Wisconsin-Madison, Madison, USA.

## Abstract

In visual cognitive neuroscience, there are two main theories about the function of the ventral visual processing stream. One suggests that it serves to classify objects (*classification hypothesis*); the other suggests that it generates intermediate representations from which people can generate verbal descriptions, actions, and other kinds of information (*distributed semantic hypothesis*). To adjudicate these, we trained two deep convolutional AlexNet models on 330,000 images belonging to 86 categories, representing the intersection of Ecoset images and the semantic norms collected by the Leuven group. One model was trained to produce category labels (*classification hypothesis*), the other to generate all of an item's semantic features (*distributed semantic hypothesis*). The two models learned very different representational geometries throughout the network. We also estimated the human semantic structure of the 86 classes by using a triadic comparison task. The representations acquired by the feature-generating model aligned better with human-perceived similarities amongst images, and better predicted human judgments in a triadic comparison task. The results thus support (*distributed semantic hypothesis*).

## 1 Introduction

Theories about human visual object recognition have generally oriented around two somewhat different views. The first proposes that the recognition system serves to match a perceived item to one of several stored object categories(Riesenhuber & Poggio, 2000; Serre et al., 2007). such matching can then index into the human semantic system to allow for retrieval of other information about the perceived item(Jolicoeur et al., 1984; Coltheart, 2004). On this *classification hypothesis*, visual object recognition is contained solely within the visual system and provides access to a separate, potentially amodal semantic store. The second perspective proposes that visual object recognition is not separate from, but is deeply influenced by, semantic processing. On this *distributed semantic hypothesis*, object recognition does not function to match a visually-perceived object to a discrete category representation. Instead, visual object processing generates a distributed, cross-modal representation that expresses the perceived item's semantic or conceptual similarities to other known items(Rogers et al., 2004). As a consequence, the representational structure expressed within recognition systems can be influenced, not only by visual structure, but also by the rich multi-modal semantic knowledge the perceiver possesses about the object.

The current paper assesses which view provides a better account of the similarities that people discern amongst photographs of objects, by training models with identical encoding architectures on identical distributions of images, but with different objectives corresponding to the two different views. We first consider whether the objective matters, showing that the two model variants learn different representational geometries that are most profound in deeper layers but persist all the way to the shallowest layers. We then consider which variant provides a better account of

human-perceived similarities, by considering how well model-based representations explain human similarity judgements, and how well model representational structure aligns with structure derived from human judgements.

**Relationship to prior work.** A broad range of recent work has considered how the internal representations acquired by contemporary deep vision models can help to understand aspects of human visual perception and/or the neural encoding of visual object information. Yamins et al. (2014) has closely considered how the behavior of deep convolutional image classifiers relate to human behavior, and how such models may explain fine-grained organization of feature-detectors within primate visual cortex when constrained with additional losses(Margalit et al., 2023). Muttenthaler et al. (2023) have considered how representations in deep image classifiers can be directly regularized to show better accordance with human similarity judgements. Fel et al. (2022) have showed that deep image classifiers can be regularised with the features that humans pay attention to while perceiving images. Mehrer et al. (2021) showed that DCNNs better capture patterns of human behavior when trained on more representative corpora of images than the usual ImageNet dataset(Deng et al., 2009) .The current work contributes to these efforts by considering an apples-to-apples comparison of models matched in all aspects except for the two losses considered, each corresponding to a different cognitive hypothesis about the functioning of the visual object recognition system.

## 2 DATASET

The dataset used to train our models was constrained by two goals. First, because we aim to understand human-perceived similarities amongst photographs, we wanted it to include a representative and ecologically realistic distribution of images. We therefore began with the ECOSET(Mehrer et al., 2021), a compilation of images belonging to 565 basic-level object categories that arguably represent the distribution of concepts with which people are typically familiar. Second, we wished to compare models that classify photographs by assigning them to the appropriate basic-level category with models that instead learn to generate a distributed semantic representation of the items. For this, we employed the *Leuven norms*, a dataset of semantic norms collected by the Leuven group(De Deyne et al., 2008), which indicates which of 2057 propositional features (e.g. "has wings", "is big," "can move," etc) are commonly judged to be true of 300 different living and nonliving categories. For such feature vectors, the feature overlap between two concepts provides an indication of their semantic relatedness(McRae et al., 2005), so that binary feature vectors encoding the norm data can serve as proxies for distributed output representations that encode semantic similarity structure. To create the dataset for this study, we therefore took the intersection of concepts appearing in both the ECOSET and the Leuven norms—a set that includes 330,000 images, each depicting an item belonging to one of 86 possible basic-level categories, and with each such category associated with a binary semantic feature vector containing 2057 elements(1). We will call this the ECOSET-86.

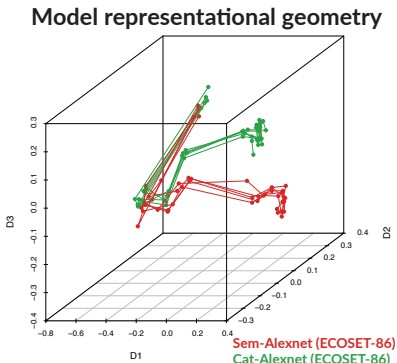

Figure 1: Second-order RDM analysis Mehrer et al. (2020) to visualise representational geometry learned by 5 instances of a model variant (Cat-Alexnet and Sem-Alexnet). Representational geometries of all instances within a model variant seem to be quite similar. Even though layers that are deeper and densely connected exhibit a stark contrast in their geometries, differences are observable across all layers, indicating that the distinct objectives of the model variants lead to systematically divergent representational geometries throughout the network.

## 3 ASSESSING

### WHETHER THE OBJECTIVE MATTERS.

Study 1 assessed whether a DCNN will acquire similar or different internal representations when trained with losses corresponding to the two hypotheses. Specificaly, *categorization* models learned to generate a one-hot basic-level category label for each image, while *distributed semantic* models learned to generate all of the semantic features associated with an image. Each model type adopted the same encoding architecture, viewed the same distribution of images, and was trained to the same performance criterion. We then assessed, at each model layer, how acquired representations compared within and between model types.

Models were variants of the AlexNet architecture (Krizhevsky et al., 2017) trained to categorize using standard cross-entropy loss (Cat-Alexnet (Table 2)) or to generate semantic features using an independent binary cross-entropy loss on each output unit (Sem-Alexnet (Table 3)). Model accuracy was measured as the proportion of times (a) the most active label was the correct label (for Cat-Alexnet) or (b) the output vector was more similar to the target vector than any other vector(for Sem-Alexnet). Five instances of each variant, differing only in initial random weight configuration, were trained to the same accuracy criterion of 70%. We then evaluated similarity of the representational geometries acquired at each model layer within and between model variants, following the procedure described by (Mehrer et al., 2020): at each layer we computed pairwise cosine distances in the elicited representations for a large test set of photographs. For each pair of layers across all models, we then computed the pearson correlation in the resulting distance matrices—producing a layer-to-layer distance matrix whose entries indicate dissimilarity in the representational geometries acquired by each model layer. Finally, we reduced this matrix to three components using classical multidimensional scaling.

Results are shown in Figure 1. Layers within a model instance are connected by a line; the two model variants (categorization vs distributed semantic) are indicated with different colors; and distances within the 3D embedding indicate dissimilarity of representational geometries between layers. These tend to be quite similar within variant, dissimilar across variants at each layer. The difference is most obvious in deeper densely-connected layers (well-separated green vs red points toward the right) but is discernable through all layers, even to the most shallow. Thus the different objectives produce systematically different representational geometries throughout the network.

Study 2 assessed whether the difference in learned representations would lead to differences in the ability of the models to predict human behavior on a triplet judgement task.

**Triadic comparison task** Our primary objective is to test the representational alignment of different model-based visual-semantic representations with human perceived visual-semantic representations. In pursuit of this, we require a means of assessing human semantic representations for our objects of interest. Here, we focus on naturalistic images of a set of inanimate and animate concepts. We adopt a method used in several recent studies investigating human conceptual structure, namely triadic similarity judgements Hebart et al. (2023); Suresh et al. (2023b;a); Mukherjee & Rogers (under revision) (Figure 2A). This entails presenting participants with sets of image triplets and instructing them to discern which among the two option images bears greater resemblance to a third reference image. By collecting enough image triplet judgements, we estimate a latent embedding (Figure 2B) for each of the unique images that participants are shown during the experiment Jamieson et al. (2015); Hornsby & Love (2020); Sievert et al. (2023). These embedding can then be leveraged to estimate a conceptual space that best predicts the judgements that humans made during the task. Below, we describe the procedure for our human experiment.

**Stimuli.** Of the 86 categories, 46 were animate and 40 were were inanimate concepts. Within the animate domain, images belonged to parent categories like Mammals, Insects, and Reptiles. Within the inanimate domain, images belonged to parent categories like Household Objects, Instruments, Vehicles, and Tools. Each category had a total of 3 images resulting in a total of 258 images.

**Participants.** We recruited 91 participants on Prolific to complete our task (Palan & Schitter, 2018). Each participant provided informed consent in compliance with our Institutional IRB.

**Procedure.** On each trial, participants viewed a target image displayed above two option images, and were instructed to choose via mouse click or arrow keys which of the two option images was

most similar to the target image. Each participant completed 570 trials. On 520 of the trials, the triplet on each trial was sampled randomly with uniform probability from the space of all possible triplets. The remaining 50 trials (randomly dispersed during the experiment) constituted 'validation triplets'. Each participant saw 50 'validation trials' from a curated pool of 258 triplets. The study yielded a total of 49970 judgments, an order of magnitude larger than the minimal needed to estimate an accurate 5-D embedding from random sampling according to estimates of sample complexity in this task (Jamieson et al., 2015; Sievert et al., 2023). 80 % of the non-validation judgments were used to estimate a 5-D embedding that minimized the crowd-kernel triplet loss on the training set of triplet judgement trials (Tamuz et al., 2011). The resulting embedding was then tested by assessing its accuracy in predicting human judgments on the non-validation held-out data. The final embeddings predicted human decisions on held-out triplets with 75% accuracy.

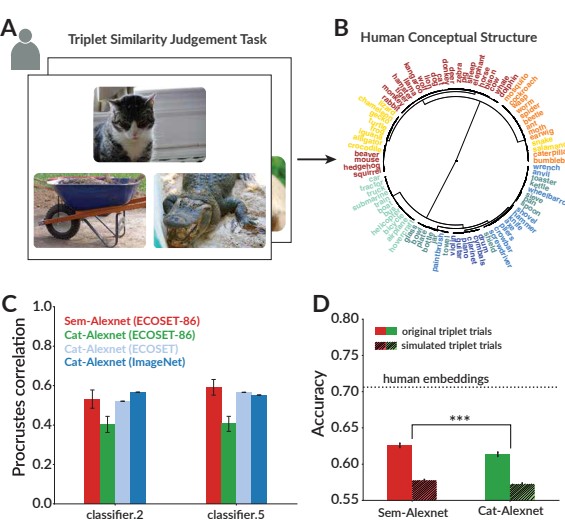

**Model evaluation.** For a direct comparison between human judgments and our model's predictions, we expose our trained models to the same set of images that were used in the triplet judgement task conducted on human participants. We used activations from intermediate layers that were present in Sem-Alexnet, Cat-Alexnet, and Alexnet pre-trained on ImageNet and ECOSET and investigated differences in learned representational geometries. To assess the structural coherence of model representations with human embeddings, we computed the the average parent category-wise Procrustes correlation of the embeddings of the 258 images derived from humans and models . This metric, analogous to Pearson *r*, indicates the extent to which variations in pairwise distances from one representational space are reflected in the other. To emulate human decision-making during each trial using the model, we computed the cosine similarity between the hidden layer representations of the images shown in the trial, thereby determining the model's choice and using accuracy as a metric of human-model alignment.

Figure 2: **(A)** Human triplet judgement task. **(B)** Hierarchical cluster plot of the 5-D human embedding derived from triplet judgement trials.**(C)** Procrustes correlations for different model layers with human judgement derived embeddings. The last 2 fully-connected dense layers of AlexNet are presented here. Sem-Alexnet at layer 'classifier.5', the deepest layer, is most aligned with human representations. **(D)** Accuracy at predicting human triplet judgements task with original triplets seen by human participants and swapped triplets with different images of the same parent categories. Sem-Alexnet is more accurate at predicting human judgements.

THE TRAINING OBJECTIVE INFLUENCES HUMAN-MODEL ALIGNMENT

*Procrustes alignment.* We find that the highest Procrustes correlation between model and human representations is achieved by the Sem-AlexNet in the deepest layer (classifier.5) that is common to all the models ( Figure 2C) . This shows that models trained on semantic feature elicitation are better aligned with human representations. It is widely believed that the model performance improves as the training dataset size increases and becomes more diverse Lei et al. (2018). To see if using an AlexNet trained on datasets that were diverse and contained more image classes improved human-model alignment, we also compared the alignment of our trained models to an AlexNet pre-trained on the entirety of Ecoset and on ImageNet. We still find that the highest Procrustes correlation is achieved by classifier.5 layer of the Sem-AlexNet which is trained on only $\sim$ 20-25% of the images (330,000) compared to ECOSET (1.5 million), and Imagenet-1k (1.2 million) images. This suggests that a training objective that is centered around human-generated semantic features allows vision models to acquire more human-like semantic

representations relative to models trained on categorization objectives. These results thus lends support to the **distributed semantic hypothesis**.

*Triplet judgement prediction.* In addition to investigating the alignment of model-human representational geometries, we were also interested in how accurate *judgments* that were based in model representations were. We specifically investigated how well models could predict human judgements on the 258 validation triplets. The 258 triplets belonged to one of three types based on how the the three images in the triplet were sampled. A third of the validation triplets were sampled such that all the three images belonged to the same parent category (eg. (cat, dog, zebra) are all mammals), a third were sampled such that two (including the anchor) belonged to the same parent category and one belonged to a different parent category (eg. (cat, dog, ant) because cat and dog are mammals and an ant is an insect), and a third were sampled such that all the three images belonged to different parent categories (eg. (cat, ant, helicopter)). We simulated the triplet judgement task using cosine distance between images from the intermediate layer activations of the model. The image that was closer in representational space to the reference image was chosen as the 'answer' on each trial. We also measured the human accuracy using the human estimated 5-D embeddings. We found that the highest triplet prediction was achieved by classifier.5 layer of the Sem-AlexNet on all three types of triplets, being the only model to reach human levels of accuracy, computed using human embedding, in two of the three triplet types and even exceeding the human accuracy when all the three images in a triplet belong to the same parent category. Predicting accuracy using a mixed-effect logistic regression model[1] also revealed that model type (Sem-Alexnet vs. Cat-Alexnet) was a significant predictor with Sem-Alexnet having the higher accuracy ($\beta$ = -0.05, $SE = 0.02$, $p < .001$). These results (Figure 2D, Figure 3) suggest that Sem-Alexnet was more aligned with humans on the triplet decision task relative to Cat-Alexnet providing additional support to the **distributed semantic hypothesis**.

THE TRAINING OBJECTIVE MAKES THE MODEL SENSITIVE TO IMAGE-SPECIFIC INFORMATION

People might gauge similarity between a triplet of images not simply using category information but visual features of the images. Is Sem-Alexnet more sensitive to this information relative to Cat-Alexnet? To test this, we compare each models' triplet judgement performance on true triplet sets shown to participants during the human experiment and on simulated 'swapped' triplet sets, where the categories in the triplet set were kept the same but the images were swapped to one of the other images belonging to each category. In other words, if the original triplet was (cat1, dog1, zebra1), we would use (cat2, dog2, zebra2) to simulate a swapped triplet. We swap the images 100 times for each triplet. We then compute the means of each of the 100 runs to the accuracy obtained when using real images. We find that Sem-Alexnet is more sensitive to image-specific properties because it is more accurate while simulating triplets than Cat-Alexnet (Figure 2D). This was supported by a significant model type $\times$ trial type (original vs. simulated) interaction effect in our mixed-effect logistic regression model ($\beta$ = .03, $SE$=.02, $p < .05$). This effect is driven by triplets of the type where all the triplets belong to the same parent category. This suggests that training models to elicit semantic features leads to learning within-category structure that is more aligned with that of humans.

## 4 DISCUSSION

While numerous models addressing visual semantics and visual perception have been proposed, our study distinguishes itself in its unique comparative methodology. To our knowledge, this is the first investigation that contrasts representations across varied models under equitable conditions—specifically,ensuring that the models are trained on identical datasets and achieve comparable performance metrics (e.g., classification accuracy). Moreover no prior work has compared image classifiers to alternative models aligning with the *distributed semantic hypothesis*–that is, models in which representations in ventral stream are partially shaped by linguistic and/or semantic knowledge. Furthermore, our study would be the first to collect a large dataset of human similarity judgments to evaluate model-human representational alignment complementing earlier such efforts Hebart et al. (2022). This dataset stands to benefit not only cognitive scientists but also computer scientists with interests in human-AI alignment. From a

---

[1] refer to Appendix for statistical model details

pragmatic standpoint, our models hold potential in the domain of medical research, particularly in the study of neurodegenerative disorders such as semantic dementia (Rogers et al., 2004) . By being the targets of controlled perturbation studies, these models stand to support controlled simulations and could elucidate why certain therapeutic interventions exhibit heightened efficacy at specific junctures in a disorder's progression — a revelation of paramount significance to the medical community.

ACKNOWLEDGMENTS

Blinded for review

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

## A  APPENDIX

Table 1: The parent categories of all the 86 categories present in our dataset

| Parent Category | Category |
|---|---|
| Mammals | cat, donkey, pig, dog, dolphin, zebra, sheep, hamster, tiger, whale, mouse, beaver, wolf, kangaroo, rabbit, monkey, hedgehog, horse, squirrel, bison, cow, deer, lion, llama, elephant |
| Reptiles | turtle, chameleon, crocodile, iguana, lizard, gecko, alligator, snake, frog, salamander |
| Insects | ant, wasp, bumblebee, mosquito, moth, beetle, caterpillar, spider, cockroach, earwig, worm |
| Household Objects | paintbrush, shield, toaster, jar, plate, glass, kettle, bowl, sieve, towel, bottle, pan, spoon, wheelbarrow |
| Instruments | piano, violin, cymbals, clarinet, drum, guitar |
| Vehicles | car, tractor, hovercraft, boat, helicopter, bus, submarine, truck, airplane, train, bicycle |
| Tools | hammer, screwdriver, shovel, wrench, anvil, pliers, knife, axe, crowbar |

| Block | Layer | Config |
|---|---|---|
| Features | Conv2d | 3, 64, kernel_size=(11, 11), stride=(4, 4), padding=(2, 2) |
| Features | ReLU | inplace=True |
| Features | MaxPool2d | kernel_size=3, stride=2, padding=0 |
| Features | Conv2d | 64, 192, kernel_size=(5, 5), stride=(1, 1), padding=(2, 2) |
| Features | ReLU | inplace=True |
| Features | MaxPool2d | kernel_size=3, stride=2, padding=0 |
| Features | Conv2d | 192, 384, kernel_size=(3, 3), stride=(1, 1), padding=(1, 1) |
| Features | ReLU | inplace=True |
| Features | Conv2d | 384, 256, kernel_size=(3, 3), stride=(1, 1), padding=(1, 1) |
| Features | ReLU | inplace=True |
| Features | Conv2d | 256, 256, kernel_size=(3, 3), stride=(1, 1), padding=(1, 1) |
| Features | ReLU | inplace=True |
| Features | MaxPool2d | kernel_size=3, stride=2, padding=0 |
| Features | AdaptiveAvgPool2d | output_size=(6, 6) |
| Classifier | Dropout | p=0.5 |
| Classifier | Linear | in_features=9216, out_features=4096 |
| Classifier | ReLU | inplace=True |
| Classifier | Dropout | p=0.5 |
| Classifier | Linear | in_features=4096, out_features=4096 |
| Classifier | ReLU | inplace=True |
| Classifier | Linear | in_features=4096, out_features=2000 |
| Classifier | ReLU | inplace=True |
| Classifier | Linear | in_features=2000, out_features=500 |
| Classifier | ReLU | inplace=True |
| Classifier | Linear | in_features=500, out_features=100 |
| Classifier | ReLU | inplace=True |
| Classifier | Linear | in_features=100, out_features=86 |

Table 2: Cat-AlexNet Architecture

## DETAILS ON FITTING LOGISTIC REGRESSION MODELS

In this study, the mixed-effects logistic regression models were fit using the lme4 package in R, specifically version 1.1.30. The bobyqa optimizer was utilized for the fitting of these mixed-effects models.

**Sem-Alexnet is more accurate at predicting human judgements and it is also more accurate when the judgements are predicted using original images vs swaped images** We fit a model predicting the accuracy of a triplet judgement from the model (Sem-Alexnet or Cat-Alexnet), the type of image (Real vs swapped), and their interaction while controlling for the layer (features.1 ,features.4, features.7, features.9, features.11). We included random intercepts for the type of triplet (all within parent category, two within parent category, and all different parent category) and the Model (Sem-Alexnet or Cat-Alexnet).

| Block | Layer | Config |
|---|---|---|
| Features | Conv2d | 3, 64, kernel_size=(11, 11), stride=(4, 4), padding=(2, 2) |
| Features | ReLU | inplace=True |
| Features | MaxPool2d | kernel_size=3, stride=2, padding=0 |
| Features | Conv2d | 64, 192, kernel_size=(5, 5), stride=(1, 1), padding=(2, 2) |
| Features | ReLU | inplace=True |
| Features | MaxPool2d | kernel_size=3, stride=2, padding=0 |
| Features | Conv2d | 192, 384, kernel_size=(3, 3), stride=(1, 1), padding=(1, 1) |
| Features | ReLU | inplace=True |
| Features | Conv2d | 384, 256, kernel_size=(3, 3), stride=(1, 1), padding=(1, 1) |
| Features | ReLU | inplace=True |
| Features | Conv2d | 256, 256, kernel_size=(3, 3), stride=(1, 1), padding=(1, 1) |
| Features | ReLU | inplace=True |
| Features | MaxPool2d | kernel_size=3, stride=2, padding=0 |
| Features | AdaptiveAvgPool2d | output_size=(6, 6) |
| Classifier | Dropout | p=0.5 |
| Classifier | Linear | in_features=9216, out_features=4096 |
| Classifier | ReLU | inplace=True |
| Classifier | Dropout | p=0.5 |
| Classifier | Linear | in_features=4096, out_features=4096 |
| Classifier | ReLU | inplace=True |
| Classifier | Linear | in_features=4096, out_features=2000 |
| Classifier | ReLU | inplace=True |
| Classifier | Linear | in_features=2000, out_features=500 |
| Classifier | ReLU | inplace=True |
| Classifier | Linear | in_features=500, out_features=100 |
| Classifier | ReLU | inplace=True |
| Classifier | Linear | in_features=100, out_features=86 |

Table 3: Sem-AlexNet Architecture

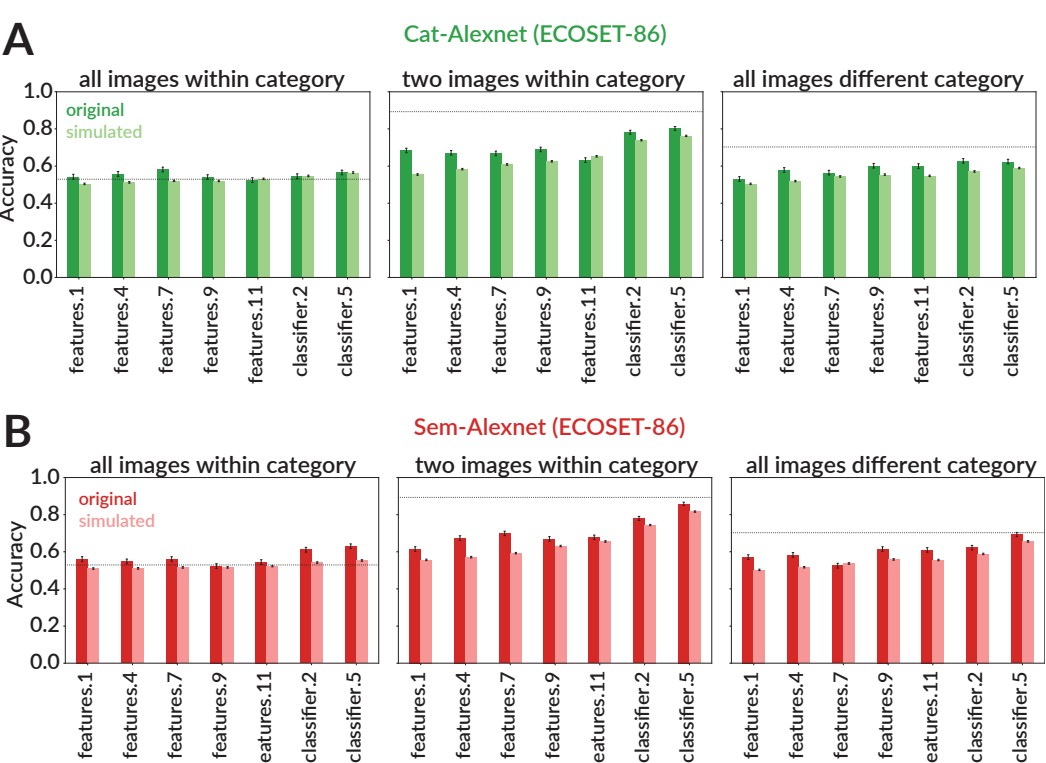

Figure 3: Model accuracy in predicting human triplet judgements for original triplet trials and simulated trials with swapped images. Sem-Alexnet (**B**) was generally more aligned with human judgements than Cat-Alexnet (**A**) with later layers in both models showing generally higher alignment for all 3 triplet trial types. Dashed lines indicate prediction accuracy from human judgement-derived embeddings.

