# OpenReview forum: "Categories vs Semantic Features: What shape the similarities people discern in photographs of objects?"
_ICLR.cc/2024/Workshop/Re-Align — ICLR 2024 Workshop Re-Align Poster_

### Official Review · Reviewer_17rU · 2024-02-19

**Rating:** 2
**Fit:** 3
**Confidence:** 2

**Workshop Review:**

**Summary:** \
In the study, the authors have created a novel dataset (ECOSET-86; 86 classes and 333,000 images), being the intersection of the widely used ECOSET and the lesser-known Leuven Norms. A less-known dataset consisted of semantic feature vectors (2057 elements) for each image. This dataset is used to train two AlexNet Models (Cat-AlexNet; Sem-AlexNet) with which the authors aim to test whether humans utilize objects/categories or semantic features during visual perception. First, the authors demonstrate that each network learns unique representational geometries, making them suitable models to test the hypothesis. This is followed by testing the alignment of these models with human object recognition behavior in the form of a triadic comparison task. Their results show that Sem-AlexNet seems to be more aligned with human behavioral judgments indicating that visual perception

---

**Clarity:** good \
**Correctness:** good \
**Novelty:** good \
**Interest to the community:** good

---

**Strong points:**
- **Well-written manuscript** (Some typos and missing words)
- **Good clarity:** The novel dataset is well explained and seems to be a great addition to the field to test the hypothesis. The models are also well described (although there seems to be a typo in the supplementary figure stating the SEM-AlexNet model). The used methods seem appropriate, well described, and could be replicated. The results are stated clearly, offer new insights, and are interpreted and discussed in a balanced way.
- **Good Visualizations**: The Figures in the paper clearly state the results, seem to be correct, and support the findings as well.
The research done in this paper is relevant and at the core of the workshop. The dataset seems to be an interesting venue to test human-AI alignment.

**Weak points:**
- The supplementary figure 3 for the Sem-AlexNet Architecture seems to have a typo as it states out_features=86 (the same as the object classes in ECOSET-86) but it should be 2057 as this is the dimension of the semantic feature vector.
- The authors test for the effect of dataset diversity in the training set by also including AlexNets trained on the full ECOSET and ImageNet. They do not control the number of classes that they use. While indeed ECOSET-86 has way fewer images and SEM-AlexNet an even slightly higher correlation with human behavior this might be because SEM-AlexNet has the highest number of output classes (2057); Cat-AlexNet (86); ECOSET (353) and ImageNet(1k classes). I think this would be a great addition to the paper to test the effect of the number of output classes for the SEM-AlexNet to ensure that the effect comes from semantic information and not just the amount of output classes.

---

**Questions:**

Question 1: Can you ensure that the different model representational geometries go back to the semantic information and are not just a result of a more diverse (larger number) of output classes? \
Question 2: Is the number of output classes wrongly stated in the supplementary Figure of SEM-AlexNet?

**Reason For Not Giving Higher Score:**

N/A

**Reason For Not Giving Lower Score:**

Reason 1: Well-written manuscript, using well-established methods to answer and relevant questions to the community. \
Reason 2: The novel dataset seems to be an interesting new venue to test human-AI alignment for visual perception beyond object labels which can benefit the community.

**Reviewer Domain:**

cognitive science

---

### Official Review · Reviewer_fdJJ · 2024-02-23
**Solid experiments supporting a hypothesis for ventral visual stream function; clarity could be improved**

**Rating:** 2
**Fit:** 3
**Confidence:** 2

**Workshop Review:**

Clarity & correctness: The point of the paper and writing style are clear. Experiments are mostly described thoroughly, though some clarifications could be made in the text to improve paper quality:
1. Training details: What is the 70% accuracy threshold evaluated over (and why not train for n epochs or until val accuracy saturates)? What is the train/val/test split?
2. What is the "classical multidimensional scaling" used to visualize the distance matrix?
3. Why learn five dimensions for the human representation? If this is justified in Jamieson et al. or Sievert et al., it would still be useful to briefly justify the low dimensionality. Experimentally, learning a >5 dimensional embedding and showing that it does not encode much additional variance would be quite convincing.
4. My understanding of Procrustes analysis is that it can compare two shapes (like distributions) by performing a set of transformations. I may just not be familiar with the correlation part of the analysis, but it is not clear to me what a correlation coefficient has to do with the shape-transformation angle that Procrustes analysis usually takes. It would be helpful to provide a citation or more rigorous formulation for how this correlation is calculated.
5. While I appreciate the citations pointing towards one hypothesis or the other, it is not clear to me that these two hypotheses are the main camps that are at odds with each other. Are there no other major theories sharing this problem space?
6. There are some typos and grammatical errors, but they are not too distracting - I would still suggest at least doing a pass to smooth over spellings and sentence readability.

The experimental design seems sound to me. Some design decisions need to be explained (see above) for thoroughness.

Novelty & interest to community: This paper provides a new way to computationally support a hypothesis for the function of the ventral visual pathway. Perhaps the learned embeddings themselves can be useful to the ML/cog-sci communities as well as the conclusions drawn from the work.

**Reason For Not Giving Higher Score:**

While the experiments are interesting, the results are not very extensive or conclusive. The bar plots in Figure 2 suggest support for the distributed semantic hypothesis, but for this to be the defining finding of this paper I would have liked to see more thorough comparisons across backbones trained on ECOSET-86 and compared to more non-ECOSET-86 datasets.

**Reason For Not Giving Lower Score:**

As detailed in the review, this paper meets the criteria for the workshop.

**Reviewer Domain:**

machine learning

---

### Official Review · Reviewer_u8LM · 2024-02-26
**Interesting assessment of theories of visual perception through the lens of model-human alignment.**

**Rating:** 2
**Fit:** 3
**Confidence:** 3

**Workshop Review:**

**Summary:** This paper aims to assess two theories regarding the function of the ventral visual: the *classification hypothesis* (that human recognition simply groups items into categories) and the *semantic hypothesis* (that the recognition system generates detailed semantically-informed representations). The papers examines these theories by training AlexNet models using training objectives that are aligned with these two hypotheses, and then testing which models are better aligned with human similarity judgements; they find better alignment with the feature-generating model, thus supporting the semantic hypothesis.

**Strengths:**
- The research question is well-formulated, and of interest to both cognitive science and machine learning researchers.
- The results around model representational geometry (Fig 1) are particularly interesting, and the trend regarding greater differences in deeper layers makes intuitive sense given that the training objective is the main difference between the two models.
- The main finding -- that the feature generating model aligns better with human judgements -- is well-supported by the experiments.

**Weaknesses**:
- The claims regarding the novelty of the collected dataset are not supported. The authors collect a dataset of image triplets with human similarity judgements, and state that they are  "the first to collect a large dataset of human similarity judgements to evaluate model-human representational alignment". However, to my knowledge, there are at least three existing, large datasets of human similarity judgements on image triplets: BAPPS [1], THINGS [2], and NIGHTS [3], all of which have been used in machine learning literature to evaluate model-human alignment. The authors cite [2] however do not explain how their dataset differs. To support statements of the dataset's novelty, they should cite existing datasets in the related works section and explain how theirs is unique.
- In plots assessing human-model alignment, the baseline is denoted by the accuracy of embeddings that predict human decisions. However this leads to results, such as in Fig. 3, where some models appear to achieve higher alignment than the human embeddings. It seems a bit strange for any model to outperform the human metric on a human alignment task -- it may be more informative for the authors to instead (or additionally) plot the inter-rater agreement as a noise ceiling.
- It may not be immediately clear to all readers why the triplet judgement task in particular is a good test of model-human representational alignment. The authors explain this well in the Section 3, but it may be helpful to include some of this reasoning in the introduction.

**Questions:**
- How much does the choice of model architecture impact these findings? Is it possible that different trends may emerge for larger models?
- It would be interesting to see if these model-human alignment results are consistent across different similarity judgement datasets.

**References**
[1] R. Zhang, P. Isola, A. A. Efros, E. Shechtman, O. Wang.  The Unreasonable Effectiveness of
Deep Features as a Perceptual Metric. In CVPR, 2018.
[2] Martin N Hebart, Oliver Contier, Lina Teichmann, Adam H Rockter, Charles Y Zheng, Alexis Kidder, Anna Corriveau, Maryam Vaziri-Pashkam, Chris I Baker (2023) THINGS-data, a multimodal collection of large-scale datasets for investigating object representations in human brain and behavior eLife 12:e82580.
[3] Fu, S., Tamir, N., Sundaram, S., Chai, L., Zhang, R., Dekel, T., & Isola, P. (2023). DreamSim: Learning New Dimensions of Human Visual Similarity using Synthetic Data. NeurIPS (2023).

**Reason For Not Giving Higher Score:**

The results are interesting, and the methodology is sound, however it is unclear if they would generalize across different datasets and model architectures. Additionally, more detailed discussion of relation to prior work is needed.

**Reason For Not Giving Lower Score:**

This paper examines a meaningful, significant question about human visual object recognition, with interesting parallels to existing work on machine learning on model-human representational alignment. It aligns well with the goals of the workshop and should spark interesting discussions between cognitive scientists and machine learning researchers.

**Reviewer Domain:**

machine learning

---

### Decision · Program_Chairs · 2024-03-02

Accept (Poster)